# Vertebral Osteosarcoma in Two Cats—Diagnosis, Treatment, and Outcome

**DOI:** 10.3390/ani13223478

**Published:** 2023-11-10

**Authors:** Koen Maurits Santifort, Martijn Beukers, Arno Roos, Benjamin van Rijswoud, Nadine Meertens, Klaas Peperkamp, Ron Ben-Amotz, Niklas Bergknut

**Affiliations:** 1IVC Evidensia Small Animal Referral Hospital Arnhem, 6825 MB Arnhem, The Netherlands; 2IVC Evidensia Small Animal Referral Hospital Hart van Brabant, 5144 AM Waalwijk, The Netherlands; 3IVC Evidensia Small Animal Referral Hospital Nieuwegein, 3433 NP Nieuwegein, The Netherlands; 4Department of Pathology, Royal GD, 7418 EZ Deventer, The Netherlands

**Keywords:** osteosarcoma, feline, outcome, treatment, chemotherapy, surgery

## Abstract

**Simple Summary:**

Vertebral osteosarcoma is infrequently reported in cats. In this case report, we describe the diagnosis, treatment, and outcome of vertebral osteosarcoma in two cats. In both cases, preliminary diagnosis was based on diagnostic imaging findings. Treatment involved surgery with the aim of gross resection and post-surgical chemotherapy. Confirmation of the diagnosis was achieved by a histopathological examination of surgical biopsies. Short-term and medium- to long-term outcomes were excellent in both cases. This case report highlights the possibility of good outcomes after the surgical treatment of feline vertebral osteosarcoma supplemented with post-surgical chemotherapy.

**Abstract:**

In this case report, we describe the diagnosis, treatment, and outcome of two feline cases of vertebral osteosarcoma. Case 1: A 6-year-old female neutered domestic longhaired cat was presented with progressive paraparesis, ataxia, and spinal hyperesthesia. MRI of the thoracolumbar spinal cord and vertebral column revealed a strongly contrast-enhancing mass lesion originating from the dorsal lamina and spinous process of T13. The lesion caused extradural compression of the spinal cord. Surgical debulking was performed, and the histopathological evaluation of surgical biopsies was consistent with vertebral osteosarcoma. The cat was paraplegic with intact nociception post-surgery. Subsequently, the cat recovered ambulation while remaining mildly ataxic and paraparetic at long-term follow-up. Post-operative chemotherapy was started with doxorubicin. CT scans at 2, 4, 9, 13, and 20 months post-surgery showed no signs of local recurrence or metastasis. Case 2: A 15.5-year-old male neutered domestic shorthaired cat was presented with progressive paraparesis, tail paresis, and spinal hyperesthesia. Radiographs and CT scan of the lumbar vertebral column showed a large mass originating from the dorsal lamina and spinous process of L6, suggestive of neoplasia, with severe compression of the spinal cord. Surgical debulking was performed, and the histopathological evaluation was consistent with vertebral osteosarcoma. Post-operative chemotherapy was started with doxorubicin. Seven months post-surgery, the patient was neurologically normal with no signs of metastatic disease. This case report highlights the possibility of good outcomes after the surgical treatment of feline vertebral osteosarcoma supplemented with post-surgical chemotherapy.

## 1. Introduction

Compared with its canine counterpart, feline osteosarcoma (FO) is infrequently diagnosed in veterinary medicine [1,2,3,4,5,6]. FO affecting the axial skeleton, or feline vertebral osteosarcoma (FVO), is only described in detail in a few veterinary publications [7,8,9,10,11]. These publications have provided important information regarding possible outcomes of surgical and medical treatment.

According to the findings of a large FO study by Heldmann et al., FVO accounts for a large percentage of skeletal FO; 40 out of 90 (44%) skeletal FO affected the axial skeleton in that study [2].

Based on the literature, aggressive surgical treatment seems to be the most effective treatment for all types of FO, irrespective of additional treatment such as chemotherapy [1,2,3,4,5,6,7,8,9,10,11]. Mean survival times are reported to be shorter for FVO compared to appendicular FO [1,2]. Local recurrence is reported after cytoreductive surgical treatment [11]. Still, there are recent case reports that document favorable long-term outcomes for the surgical treatment (with additional chemotherapy) of FVO [7,8,9,10].

In this case report, we describe the diagnosis, treatment (including surgery and chemotherapy), and outcome of two cats with vertebral osteosarcoma.

## 2. History and Case Presentation

### 2.1. Case 1

A 6-year-old female neutered domestic shorthair cat was presented for chronic progressive paraparesis and spinal hyperesthesia. The referring veterinarian had prescribed meloxicam (0.05 mg/kg q24h per os) and gabapentin (10 mg/kg q8h per os), but this treatment had not resulted in a significant improvement. Residual signs of back pain were reported by the owner, especially when that cat was picked up. No traumatic events were reported by the owner. Upon presentation to the referral hospital, a general physical examination revealed no abnormalities. Neurological examination revealed mild, ambulatory paraparesis and proprioceptive ataxia, with delayed hopping responses of the left pelvic limb. Severe hyperesthesia was noted upon palpation of the vertebral column in the area of the thoracolumbar junction. An asymmetric (left-sided) T3-L3 myelopathy was suspected. Hematology and biochemical blood tests revealed no significant abnormalities. An MRI study of the thoracolumbosacral spinal cord and surrounding structures, including the T2-sacral vertebrae, was performed under general anesthesia. The following sequences were included: sagittal plane T2-weighted (T2W), T1W, and short-tau inversion recovery (STIR); dorsal plane 3D fast gradient echo combined with water excitation technique (FFE3D combined with WET), T1W, and T2W with fat saturation; transverse plane T2*W, T1W, T2W, and T2W with fat saturation over the region of the T12-L2; and post-contrast (IV gadolinium 0.15 mmol/kg) T1W sagittal plane, T1W transverse plane, and 3D T1W magnetization prepared—rapid gradient echo (MPRAGE) sagittal plane (and reconstructions in transverse and dorsal planes).

MRI revealed a 1.2 cm × 1 cm × 1.1 cm mass lesion centered on the dorsal lamina and left caudal articular process of T13, with disruption of the cortex. Relative to the spinal cord, the lesion was homogeneous and mildly hyperintense on T1W images, and heterogeneously hyperintense on T2W and STIR images (Figure 1). The mass showed strong, diffuse contrast enhancement, mainly in the areas of T2W hyperintensity (Figure 1). The mass had well-defined margins and caused moderate to severe extradural spinal cord compression. No intramedullary signal changes were seen. Based on these MRI findings, a primary bone tumor was suspected.

Surgical treatment was selected. General anesthesia was performed, and the cat was positioned in sternal recumbency. A dorsal midline skin incision was performed from T12 to L2. The fascia was incised just paramedial to the spinous processes on the left and right side. The spinous process of T13 was localized with fluoroscopy. The epaxial muscles were elevated from the spinous processes with a periosteal elevator and retracted laterally on both sides. Hemostasis was achieved using bipolar electrocautery. The zygapophyseal (facet) joint T13-L1 on the left was partially removed using a rongeur. The most ventral part was left in situ. The mass was directly visible between the dorsal laminae of T13-L1, and the lamina of T13 was thin. The tumor was debulked based on gross appearance. However, complete excision was deemed unlikely due to invasiveness into the surrounding bony margins of T13. Decompression was achieved and deemed satisfactory based on the visualization of the spinal cord in the expected position in the spinal canal. A thin autologous free fat graft was placed to cover the laminectomy defect. Monofilament absorbable suture materials were used to close the fascia, subcutis, and skin (intradermal). Post-operative care consisted of urinary bladder catheterization, ketamine (5 ug/kg/h) for 24 h, meloxicam (0.05 mg/kg q24h for 2 weeks), amoxicillin/clavulanic acid (15 mg/kg q12h for 5 days), and gabapentin (10 mg/kg q8h for 3 weeks). Physiotherapy was initiated within 2 weeks post-surgery. A neurological examination was performed 24 h post-surgery and revealed paraplegia without nociception. During the next 12 weeks, nociception, urinary and fecal continence, ambulation, and coordination returned progressively. Complete neurological recovery was achieved 3 months post-surgery.

The histopathological examination of surgical biopsies revealed a proliferation of polygonal to plump fusiform mesenchymal cells in clusters with the formation of osteoid in-between the cells (Figure 2). The cells had a round to oval, vesicular nucleus with occasional distinct nucleolus and a small quantity of basophilic cytoplasm. The cells showed moderate anisocytosis and anisokaryosis, with 4 mitoses per 10 high-power fields. Local multinucleated cells were present.These findings were consistent with a vertebral osteosarcoma, osteoblastic productive morphological subtype.

Chemotherapy was initiated 2 weeks post-surgery. A total of six doses of doxorubicin (1 mg/kg, intravenous) were administered every 3 weeks. Before each treatment, hematologic blood values were checked. Two months post-operatively, a CT scan was performed to monitor for local recurrence or signs of metastasis. No signs of local recurrence were found, but a small soft tissue nodule was noticed in both the left and right cranial lung lobes. Therefore, it was advised to repeat the CT scans during the course of treatment. Computed tomographic (CT) scans at 4, 9, 13, and 20 months post-surgery showed no signs of local recurrence or metastasis. The previously noted nodules were not present anymore. The cat was reported to be doing well 28 months post-surgery. No significant side effects were noticed during treatment.

### 2.2. Case 2

A 15.5-year-old male neutered domestic shorthaired cat was presented for chronic progressive lameness of the right pelvic limb over the last 4 months. During the last 2 weeks, the signs had progressed with the involvement of the left pelvic limb as well. The referring veterinarian had prescribed meloxicam (0.05 mg/kg q24h per os). However, a significant improvement was not achieved. During a follow-up neurological evaluation, proprioceptive deficits and spinal hyperesthesia were noted. Orthogonal radiographs of the lumbar vertebral column revealed a right-sided osteolytic mass lesion affecting the L6 vertebra (Figure 3). Gabapentin (10 mg/kg q12h) was added to the treatment and a CT scan was performed. The CT revealed a large (2.2 × 2.0 × 2.9 cm), expansile mass lesion in the dorsal lamina of the L6 vertebra (Figure 4). The mass also infiltrated the pedicles bilaterally (right more than left). The mass showed soft tissue attenuation centrally and a thin, mineralized peripheral rim. Its borders were mildly irregular and quite well defined. There was severe stenosis of the spinal canal, mainly at the level of the caudal part of L6. Caudally, the mass extended dorsally to L7. It did not appear to affect any part of the L7 vertebra. The soft tissue attenuating portion showed diffuse contrast enhancement. Differential diagnoses based on the CT findings mainly included a primary bone tumor such as fibrosarcoma, osteosarcoma, or chondrosarcoma. A fine-needle aspiration biopsy of the mass was attempted, but the acquired sample was non-diagnostic. Surgical treatment was planned. The dosage of gabapentin was increased (10 mg/kg q8h), and meloxicam was discontinued and replaced with prednisolone (1 mg/kg q24h) 2 weeks before surgery.

Upon presentation to the referral hospital for surgical treatment, a general physical examination revealed no abnormalities. Neurological examination revealed ambulatory paraparesis and hopping deficits of the right and left pelvic limb (right more than left). Hyperesthesia was noted upon palpation of the vertebral column in the area of L6. Hematology and biochemical blood tests revealed no significant abnormalities aside from a hematocrit of 25.7% (reference range 30.3–52.3%). Pre-operative CT revealed similar findings regarding the L6 mass lesion. Additional findings included an enlarged left adrenal gland, mild hepatomegaly, mild diffuse bronchial wall thickening, a mass lesion in the pituitary fossa measuring 5.3 mm dorsoventral height (suspected pituitary tumor), degenerative joint disease of multiple joints, and multifocal spondylosis deformans. The owners did not elect further diagnostic tests for these findings and elected to proceed with the surgical treatment for the L6 mass lesion.

A dorsal midline skin incision was performed from L4 to S1. The fascia was bilaterally incised on the midline dorsal to L5-L7. The epaxial muscles were elevated from the spinous processes with a periosteal elevator and retracted laterally on both sides. Hemostasis was achieved using bipolar electrocautery. The tumor was visualized. It had a very hard consistency on the outer aspect. The dorsal and right lateral parts of the tumor wall were removed using rongeurs. Thereafter, the soft content of the mass was removed by curettage. The compressive, thin external bony layer was easily removed using a curette and a Kerrison rongeur. The L6 spinal nerve on the left side was accidentally severed in the process. Decompression was achieved and deemed satisfactory based on the visualization of the spinal cord and nerve roots in the expected position in the spinal canal. A thin autologous free fat graft was placed to cover the laminectomy defect. A molded piece of polymethylmethacrylate (PMMA) was placed dorsally to L5-sacrum to replace the removed dorsal lamina and cover the spinal canal. Monofilament absorbable suture materials were used to close the fascia, subcutis, and skin (intradermal). Decompression was evaluated as satisfactory on post-operative CT images (Figure 4). Post-operative care consisted of urinary bladder catheterization, ketamine (5 ug/kg/h) for 24 h, prednisolone (1 mg/kg q24h for 2 weeks, tapering scheme thereafter over the next 2 months), and gabapentin (10 mg/kg q8h for 3 weeks). One day post-surgery, the cat was ambulatory paraparetic. Improvement in neurological function was noted over the next 3 months post-surgery, at which point no residual signs were noticed by the owner.

The histopathological examination of the surgical biopsies revealed a highly cellular proliferation of spindle-shaped mesenchymal cells in bundles and whorls, with scattered multinucleated giant cells (Figure 5). In between the tumor cells, a minimal quantity of osteoid matrix was present. The cells had an oval to fusiform nucleus with a coarse chromatin pattern and occasional distinct nucleolus. The cells showed mild anisocytosis and anisokaryosis. There were three mitoses per 10 high-power fields. At one location, the tumor cells formed an expansile, well-circumscribed frond and did not show infiltrative behavior in the surrounding bone tissue; therefore, the mass seemed to be relatively well delineated as far as could be judged based on the surgical biopsy. These findings were consistent with a vertebral osteosarcoma of fibroblastic morphological subtype.

The cat was referred for further treatment by the oncology department. Chemotherapy was initiated 3 weeks post-surgery. A total of four doses of doxorubicin (1 mg/kg, intravenous) were administered every 3 weeks. Before each treatment, hematologic blood values were checked. Two months post-operatively, thoracic radiographs were acquired to monitor for signs of metastasis. No abnormalities were noticed. Additionally, radiographs of the surgical site revealed no complications related to the PMMA implant. After the third dose of doxorubicin, the patient had developed a mild neutropenia of 2.15 × 10^9^/L (grade 1 neutropenia). The fourth doxorubicin treatment was postponed for 10 days, when the values were rechecked and the neutropenia was normalized to 3.81 × 10^9^/L. No other significant side effects were noticed during treatment. Seven months post-surgery, the patient was neurologically normal, with no signs of metastatic disease.

## 3. Discussion

This case report highlights the possibility of good outcomes after surgical treatment of FVO supplemented with post-surgical chemotherapy. Such positive outcomes have been documented in a few cases of FVO before [7,8,9,10]. Along with those reports, our case report provides further support for the consideration of such treatments for FVO.

Surgical treatment options of FVO include but are not limited to cytoreduction, extirpation, debulking, and vertebral replacement [1,2,6,7,8,9,10]. Depending on the exact parts of the vertebra involved, an approach commonly used is a dorsal approach for a dorsal laminectomy, with decompression of the spinal cord being achieved by the removal of the mass. Hemilaminectomy procedures as well as corpectomy with spinal stabilization procedures are reported as well. In case 1, intra-operative fluoroscopy was used to aid in the location of the spinous process of T13. Based on presurgical imaging, a modified dorsal hemilaminectomy procedure was selected. During the procedure, the ventral part of the joint facet of T13-L1 was left intact. This approach was selected in attempt to preserve the stability of the vertebral column. In case 2, one could say a modified hemilaminectomy procedure was performed as the dorsal, and the right lateral part of the tumor wall was removed. A thin remnant of the external bony layer covering and completely compressing the spinal cord was removed using a curette and a Kerrison rongeur. PMMA was placed dorsally to L5-sacrum to cover the resulting defect. The decision to do so was based on excessive exposure of the spinal cord and nerve tissue to the surrounding soft tissues without any covering of the defect. During surgery of case 2, the L6 nerve root was accidentally severed. This did not result in lasting neurological deficits as the cat was deemed neurologically normal at last follow-up (7 months post-surgery).

The use and value of radiographs, CT (with or without myelography), and MRI for the diagnosis of FO and FVO has been reported [7,8,9,10,11,12,13]. Features that are reported for osteosarcoma on radiographs include cortical lysis, new bone formation (e.g., perpendicular to cortical bone (sunburst effect)), and the presence of Codman’s triangle (lifting of the periosteum at the periphery of the lesion). Pathological fractures may be present. The differentiation of normal bone from affected, neoplastic bone can be difficult. Osteomyelitis (e.g., bacterial or fungal) may be a differential diagnosis for such findings [14,15].

Imaging characteristics of FVOs are only reported in a few case reports and described as space-occupying osteolytic lesions that cause extradural spinal cord compression [7,8,9,10]. The mineralized component of the lesions varied from none (soft tissue attenuation [7]) to mild periosteal new bone formation [8], or a small mineralized component and/or an ossified peripheral rim [9,10]. The mass in case 2 shows similar imaging characteristics compared to a case of FVO in C7 [10] and of a pelvic FO, with a peripheral ossified border [12]. In all cases in which CT and/or MRI post-contrast images were acquired, including the two cases presented here, there was homogeneous—often strong—contrast enhancement of the mass [7,8,10]. FVO shows variable signal intensities on MRI. Compared to the spinal cord, they can be iso- or hyperintense on T1W images [7,8,10]. The T2W signal intensity varied from hypointense [10] to isointense [7] and to hyperintense [8].

FVO seems to have quite well-defined margins [7,8,9,10]. In our two cases, this was also seen. Based on the fairly good prognosis after the surgical removal of the lesion, this imaging finding likely reflects the histological margination as well, though further studies are required to confirm this.

MRI is considered the most accurate imaging modality to determine tumor margins in dogs with intramedullary osteosarcoma of the appendicular skeleton [16]. In dogs, one study identified the benefits of CT for more accurate estimation of tumor length in appendicular osteosarcoma [17]. MRI was evaluated in that study as well, and although less accurate, tumor length was not underestimated in any limb. As MRI provides much more information on the spinal cord, CT provides a better evaluation of the axial skeleton than radiographs; therefore, CT and MRI should definitely be considered for the evaluation and diagnosis of FVO. Future studies on FO and FVO in particular may support the use of MRI in the diagnosis thereof.

Since osteosarcoma is much more prevalent in dogs than cats, a short discussion with the aim of comparing dogs to cats is provided here. Osteosarcoma often affects the appendicular skeleton in dogs [13,18,19]. The incidence of axial skeletal osteosarcoma is less common, with only 13–17% being of vertebral origin [18,19].

Macroscopic metastatic disease in appendicular osteosarcoma is seen in less than 15% of dogs at first presentation, but most are presumed to have undetectable micro-metastatic disease. For this reason, chemotherapy is strongly recommended and has been shown to increase survival time significantly. The median survival time (MST) after limb amputation is just 3-4 months, but with chemotherapy, post-operative MST increases to 307-366 days [19,20,21].

Metastatic disease in canine vertebral osteosarcoma seems to occur less commonly when compared to the appendicular osteosarcomas, but the MST is worse. An MST of just 42 days is reported in dogs with vertebral osteosarcoma after surgery, while the MST of surgery with chemotherapy was 82 days, and surgery with chemotherapy and radiotherapy was 261 days [22]. The local recurrence or tumor progression was the cause of death in all cases.

Appendicular osteosarcoma occurs more rarely in cats compared to dogs, but the MST is much longer depending on the anatomic location and histological grade. A low percentage of about 5–10% of the FO is reported to metastasize [9,13,23,24]. There are indications that the osteosarcoma of the humerus or scapula behaves more aggressively, similar to those in dogs [24]. The MST of appendicular osteosarcomas in cats ranges from 2 to 5 years after limb amputation as a sole treatment [24].

About 28–44% of FO in cats is of axial origin, including the skull, vertebrae, ribs, and pelvic bones [2,24,25]. In one study of 42 cases with axial skeleton involvement, vertebrae were affected in just 2 cases [25]. Still, FO is the second most common tumor affecting the spinal cord [4]. An MST of just 5-6 months is reported for FVO, with or without radiotherapy [1,2,8,9]. The low MST of feline axial osteosarcoma compared to appendicular osteosarcoma is probably due to the likelihood of local recurrence or local tumor progression.

Because of the usually low metastatic rate in cats, chemotherapy is variably recommended [8]. But the complete resection of FVO is challenging due to the tumor location and potential risk of local invasion [9,23]. For this reason, we decided to treat both cats with adjuvant chemotherapy.

Although some case reports of feline osteosarcoma that employed surgery and adjunctive chemotherapy have shown long ST and recurrence-free periods, currently, there are no studies that evaluate the role of chemotherapy and radiation in the management of FVO [2,8,26,27]. In this case report, we document two feline cases that survived for at least 28 months and 7 months, respectively.

The limitations of the current case report include the relatively limited length of follow-up for the second case (medium-term; 7 months), the lack of further characterization of tumor type and immunohistochemistry, and the lack of further diagnostic testing with regard to staging the patients. Nevertheless, this case report adds to the existing body of literature that describes the possibility of positive outcomes of the surgical treatment, supplemented with chemotherapy, of FVO.

## 4. Conclusions

Based on previous case reports and the findings in the cases reported here, the surgical treatment of feline vertebral osteosarcoma, supplemented with post-surgical chemotherapy, can result in excellent medium- to long-term outcomes.

## Figures and Tables

**Figure 1 animals-13-03478-f001:**
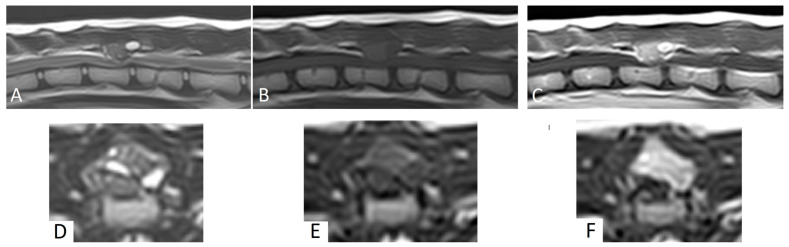
Magnetic resonance images of case 1. (**A**–**C**): Sagittal plane MRI, (**D**–**F**): transverse plane MRI. (**A**): T2W, (**B**): T1W, (**C**): T1W post-contrast, (**D**): T2W, (**E**): T1W, (**F**): T1W post-contrast. Note the mass lesion at the level of T13 with spinal cord compression.

**Figure 2 animals-13-03478-f002:**
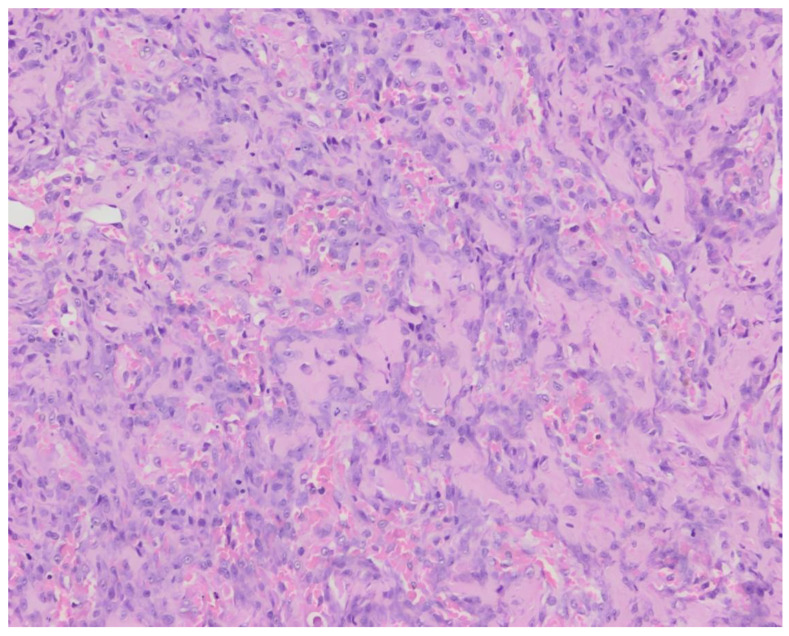
Histopathology of surgical biopsies of case 1. Hematoxylin and eosin stain, 200×. Polygonal to fusiform tumor cells surrounding eosinophilic, relatively abundant osteoid.

**Figure 3 animals-13-03478-f003:**
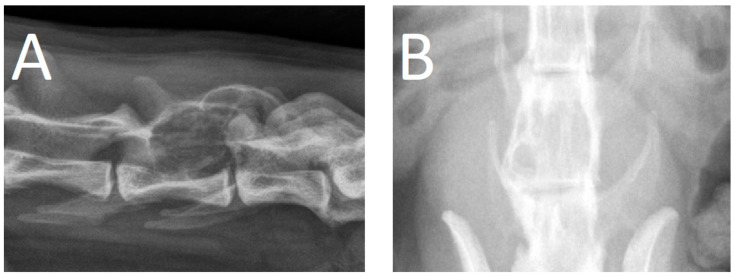
Orthogonal radiographs of case 2, provided by the referring veterinarian. (**A**): Laterolateral (right–left), (**B**): ventrodorsal. Note the osteolytic mass lesion affecting the dorsal aspect of L6.

**Figure 4 animals-13-03478-f004:**
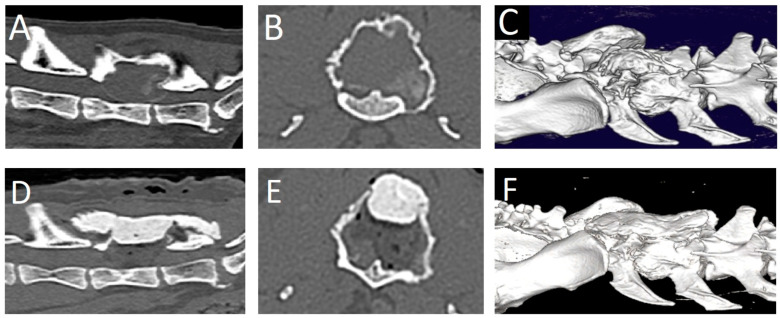
Computed tomography images of case 2. Pre-operative: (**A**): sagittal plane, (**B**): transverse plane, (**C**): 3D reconstruction. Note the irregular, lobulated mass with soft tissue attenuation in the center and a mineralized outer rim. There is severe stenosis of the spinal canal due to the mass. Post-operative: (**D**): sagittal plane, (**E**): transverse plane, (**F**): 3D reconstruction.

**Figure 5 animals-13-03478-f005:**
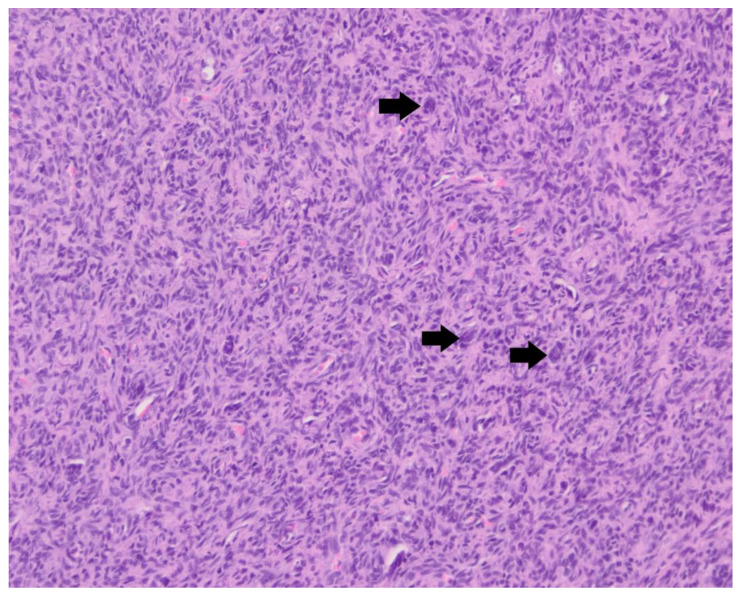
Histopathology of surgical biopsies of case 2. Hematoxylin and eosin stain, 200×. Highly cellular proliferation of spindle-shaped mesenchymal cells with scattered multinucleated giant cells (arrows highlight 3 examples thereof).

## Data Availability

All of the data are available in the present manuscript.

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
