# Peer review of "Vertebral Osteosarcoma in Two Cats—Diagnosis, Treatment, and Outcome"

_animals, 2023, doi:10.3390/ani13223478_

Round 1

Reviewer 1 Report

Comments and Suggestions for Authors

Dear Authors,

I have read your paper with great interest.

I have only few comments regarding noticed mistakes:

- line 24: you state the cat was treated with prednisolone and gabapentin, while in case description meloxicam is noted (lines 61 & 107) - please unify

- line 59: In case 1 presentation you wrote 6-y.o. domestic shorthair cat, while in Abstract (line 20) you wrote 5.5-y.o. domestic longhair cat - please unify

- line 125: doxorubicine is mentioned as chemoterapy, while in the abstract a combination of doxorubicine and pamidronic acid is noted

- lines 134-137: please delete the paragraph

Author Response

Dear reviewer,

Thank you for your review efforts and time.
Indeed, there were some mistakes in the abstract which you have correctly noted. We have made the necessary adjustments. 
The paragraph was deleted (somehow, that piece of text was incorrectly included).

Kind regards, 

Authors

Reviewer 2 Report

Comments and Suggestions for Authors

There is a difference between what is described in the Abstract, line 20, and what is described in point 2, line 59;

Point 2, line 134???

Figure 5: no arrows

Abstract, line 28 refers the use of pamidronic acid. This treatment is not mentioned in the case description, lines 125 and 126

Line 254: Bibliographical references should be added

Author Response

Dear reviewer,

Thank you for your time and effort in this review process.

We have addressed your comments, the mistakes in the abstract have been changed and the paragraph/line 134 deleted. Arrows for fig. 5 are added.

References have been added to the osteomyelitis differential sentence.

Kind regards,

Authors

Reviewer 3 Report

Comments and Suggestions for Authors

The article describes two clinical cases of vertebral osteosarcoma in two cats, analyzing them from a clinical, diagnostic, surgical and therapeutic point of view. In the first case with a long follow-up and in the second limited to only 7 months.

The introduction should be implemented with the information available on feline osteosarcoma and in particular on axial osteosarcomas.

Figure 1 D, E, F are out of focus.

Figure 2 is out of focus and looks more like a hemangiosarcoma than an osteosarcoma. Has an FVIII and/or CD31 been done to rule out hemangiosarcoma? perhaps the photo was taken in an unrepresentative and misleading place, can you take another one?

line 135 to 137 must be deleted

figure 4 A, B, D, E are out of focus

line 248 the parenthesis is not closed

Author Response

Dear reviewer,

Thank you for your time and effort in reviewing this manuscript.

We address your comments below. If you have any further concerns/comments, please let us know.
Kind regards,

Authors

The introduction should be implemented with the information available on feline osteosarcoma and in particular on axial osteosarcomas.

--> Unusually, there were three reviewers for this manuscript and the two other reviewers did not make similar remarks on the introduction. The information requested is, in part, covered in the current versions of the introduction and discussion. Should you (reviewer 3) or the editor deem it necessary nonetheless, please let us know.

Figure 1 D, E, F are out of focus.

--> The DPI of all figures has been adjusted (but the figures included in the word file are pasted in). Please consider that both the MRI and CT images in this article have a certain spatial resolution. We cannot alter the images themselves, as this would be considered a form of plagiarism. Alternatively, it would be possibly to include a larger piece of the field of view. That way, to the naked eye, the 'blurryness' would seem less, though the structures would be smaller in the image.

Figure 2 is out of focus and looks more like a hemangiosarcoma than an osteosarcoma. Has an FVIII and/or CD31 been done to rule out hemangiosarcoma? perhaps the photo was taken in an unrepresentative and misleading place, can you take another one?

--> The DPI has of this figure has been adjusted. There is osteoid matrix visible in this image. There are also areas abounding trabecular bone et cetera not included in this image. The authors feel this image is representative though, but if you deem it necessary, we will consider adding a new figure here. Please do let us know.

line 135 to 137 must be deleted

--> Indeed, sorry for this mistake. These have been deleted.

figure 4 A, B, D, E are out of focus

--> The DPI of all figures has been adjusted (but the figures included in the word file are pasted in). Please consider that both the MRI and CT images in this article have a certain spatial resolution. We cannot alter the images themselves, as this would be considered a form of plagiarism. Alternatively, it would be possibly to include a larger piece of the field of view. That way, to the naked eye, the 'blurryness' would seem less, though the structures would be smaller in the image.

line 248 the parenthesis is not closed

--> Adjusted 

Kind regards, thank you again,

Authors